# Effect of Dietary Supplementation with Mannose Oligosaccharides on the Body Condition, Lactation Performance and Their Offspring of Heat-Stressed Sows

**DOI:** 10.3390/ani12111397

**Published:** 2022-05-29

**Authors:** Ying Ren, Zibin Zheng, Taotao Wu, Long Lei, Zhengya Liu, Yuanqi Zhao, Shengjun Zhao

**Affiliations:** Hubei Key Laboratory of Animal Nutrition and Feed Science, Wuhan Polytechnic University, Wuhan 430023, China; ranee1974@163.com (Y.R.); whpuzhengzibin@163.com (Z.Z.); xiaowua2020@163.com (T.W.); leilong199202@163.com (L.L.); liuzhengyaznl@163.com (Z.L.); yqzhao09@163.com (Y.Z.)

**Keywords:** sows, mannose oligosaccharide (MOS), heat stress (HS), weight loss, reproductive performance, lactation performance

## Abstract

**Simple Summary:**

Summer heat stress (HS) seriously affects the reproductive and lactation performance of sows and the long-term development of their offspring. Mannose oligosaccharide (MOS) is widely used as an ingredient in animal feed as it can limit the colonization of enteric pathogens. However, most animal experiments that use MOS to alleviate the effects of HS are often performed with broilers. The effect of MOS in sows affected by HS is unclear. Based on our results, dietary supplementation with MOS tended to alleviate HS in sows compared with the control. However, no significant interactive effects were found.

**Abstract:**

The aim of this study was to determine the effects of dietary supplementation with mannose oligosaccharide (MOS) on the condition of the body and the reproductive and lactation performances of sows. Eighty pregnant sows were randomly assigned to four groups with a 2 × 2 factorial design: with or without MOS (1 g/kg) and with or without heat stress (HS) challenge. The temperature in the HS groups (HS and HM group) was controlled at 31.56 ± 1.22 °C, while the temperature in the active cooling (AC) groups (AC and AM group) was controlled at 23.49 ± 0.72 °C. The weight loss of sows in the AC group was significantly lower than that of sows in the HS group (*p* < 0.01). The weight and backfat thickness loss of sows supplemented with MOS displayed a downward trend. The average birth weight of the litter significantly increased in the HM group (basic diet + MOS) compared with the HS group (*p* < 0.05). The milk protein of sows significantly decreased under the HS condition at 2 and 12 h after delivery (*p* < 0.05). However, the milk immunoglobin G (IgG) of sows in the HS group increased significantly compared with that of sows in the HM group (*p* < 0.05) at 12 and 24 h after delivery. The levels of serum urea nitrogen (UREA) and glucose (GLU) decreased significantly under the HS condition (*p* < 0.05), while the level of interleukin-6 (IL-6) increased significantly under the HS condition (*p* < 0.05). Dietary supplementation with MOS also significantly reduced TNF-α under the AC conditions (*p* < 0.05). In conclusion, HS significantly affected the body condition, lactation performances and their offspring of sows. However, dietary supplementation with 1 g/kg MOS did not result in statistically significant changes.

## 1. Introduction

Pig production faces seasonal fluctuations, especially during summer heat stress (HS). HS leads to low farrowing rates of breeding sows during the summer. Based on prior studies, hot season reduces sperm motility and concentration in boars, resulting in lower farrowing rates in summer mating sows [1,2]. Sows that mate in the summer are also more prone to early pregnancy abortion than those that mate in other seasons [3]. HS increases carcass fat in the offspring of sows bred in the summer by reducing fetal muscle fiber development during pregnancy [4]. HS reduces feed intake and the growth rates of pigs in the summer [5], and leads to a reduction in milk production in lactating sows [6]. These three adverse effects induced by HS are identified as important factors that affect the pig industry [7].

Mannose oligosaccharide (MOS) is a functional oligosaccharide derived from the outer layer of the yeast cell wall and is considered as an alternative to antibiotics in animal production [8]. As a recognized prebiotic, MOS can reduce the adverse effects of HS on animals [9,10]. Supplementing 1 g/kg MOS in broiler diets for 42 days reduced serum tumor necrosis factor alpha (TNF-α) content, liver Toll-like receptor 4 (TLR4), and TNF-α mRNA abundance under HS conditions (32–33 °C) [9]. Previously, Liu revealed that under HS conditions, dietary supplementation with 0.8 ppm Se, 1% yeast nucleotides, and 0.1% MOS significantly reduced sow body weight loss (*p* = 0.037) [10]. Supplementation of MOS at 250 mg/kg in the diet was effective in improving broiler growth performance (ADG, ADFI and feed conversion ratio) [11]. Supplementing MOS in the diet improved the quality of colostrum, and promoted piglet growth performance and innate immunity in lactating and nursery piglets [8]. The increase in ADG was due to the promotion of MOS on nutrients digestibility, gut microbiology, and barrier integrity, Oxidative state and animal immunity. He et al. showed that gut microbiota composition responding to HS could be considered as biomarkers in monogastric animals [12]. MOS promotes the growth of beneficial bacteria in the digestive tract of animals and suppresses the growth of pathogenic bacteria [13,14]. Beneficial intestinal bacteria, such as *Lactobacillus* and *Bifidobacterium*, utilize MOS to produce short chain fatty acids (SCFA) and lactic acid (LA), reduce the pH value in the intestine and prevent pathogens, such as *Escherichia coli* and *Clostridium perfringens*, from adhering to the intestinal mucosa [14,15]. Studies have shown that 12 h of HS can induce an inflammatory response in porcine skeletal muscle [16]. Diet supplemented with MOS may contribute to the maintenance of intestinal integrity and nutrient digestion and absorption in the intestine after weaning [8]. As an immune regulator and prebiotic [17], MOS may alleviate the adverse reactions under HS by improving the gut microbiota and nutrient absorption of pigs. 

Extensive studies have demonstrated that dietary supplementation with MOS could improve growth performance [18], enhance the immune ability of intestinal mucosa, and inhibit the intestinal and systemic inflammatory response of weaned piglets [19]. In an HS environment, dietary MOS supplementation improved inflammatory response in broiler liver [9], and improve their growth performance, oxidation state, and integrity of the intestinal barrier [11]. Most studies on MOS in animals under HS conditions have been performed with broilers. Thus, studies on the regulatory effect of MOS in sows in an HS environment are limited. The present study sought to explore the effects of supplementing diets with MOS on the reproductive performance and lactation performance of sows to establish a theoretical basis for reducing the adverse reactions of sows under an HS environment and reduce the economic losses in the pig industry during HS conditions.

## 2. Materials and Methods

### 2.1. Animal Care

All sows were handled according to the guidelines of the Animal Care and Use Committee of Wuhan Polytechnic University. A total of 80 pregnant sows (Landrace × Large white, primiparous (*n* = 28)/multiparous (*n* = 52)) were controlled in the creditworthy farm of Wuhan Jinying Animal Husbandry Co., Ltd. (Wuhan, China). The experiment started after 100 d of gestation and lasted up to 21 d postpartum. Accordingly, the total period lasted for 36 days. After 100 days of gestation, all sows within each dietary treatment group were equally assigned to traditional delivery houses (heat stress, HS) or air conditioning delivery houses (active cooling, AC) equipped with pad-fan cooling system and shading net [20]. During the experiment, the real-time temperature and humidity of the environment were recorded every hour using an automatic temperature and humidity recorder (W-series, Wuhan, China), including daily maximum and minimum temperature (°C) and relative humidity (%). Daily temperature-humidity index (THI) was calculated using the formula recommended by the National Research Council (1971) (NRC) [21]:THI = (1.8 × *T_db_* + 32) − [(0.55 − 0.0055 × RH) × (1.8 × *T_db_* − 26.8)]
where *T_db_* is the dry bulb temperature (°C) and RH is the daily relative humidity (%). THI has been popularly used to indicate HS in animals [22,23]. The feeding trial was conducted from Day 100 of gestation until Day 21 of lactation (at weaning) at an approximately 4000-sow commercial pig farm in a subtropical monsoon climate (113.41° E, 29.58° N, Wuhan, China).

### 2.2. Experimental Design and Diet

A 2 × 2 factorial design comprising environment (HS vs. AC) and dietary treatments (basal diet vs. basal diet + additive MOS) was employed in this study. Based on blocking by fetal times, pregnant sows (*n* = 80) were randomly assigned to 4 groups with 20 sows each: (1) Basal diet in a heated environment (negative control, H); (2) Basal diet + additive MOS (1 g/kg, Phileo-lesaffre Animal Care) in a heated environment (HM); (3) Basal diet in an appropriate environment (positive control, C); and (4) Basal diet + additive MOS (1 g/kg, Phileo-lesaffre Animal Care) in an active cooling environment (CM). The dose of MOS was based on a previous study [19]. Sows were only administered 2.5–4 kg/d of feeding from entry until Day 3 after farrowing. Thereafter, they were fed *ad libitum* as previously described, and granted free access to tap water during the experiment. The basal diets administered during gestation and lactation were formulated to meet the commercial standards (Tangrenshen Group Co., Ltd., Zhuzhou, China). The ingredients and formulated dietary nutrients are shown in Table 1. Sows were fed the gestation diet before farrowing and the lactation diet after farrowing (twice per day at 8:00 and 18:00). Other breeding management was performed according to the management system of the farms. The health status of the animals was also recorded. 

### 2.3. Sample Collection

The body weight and back-fat thickness (using Renco Lean-Meater digital backfat indicator) of sows were measured upon entry to the farrowing house and at weaning. Daily feed intake, the occurrence of constipation, and the duration of parturition of sows were recorded. Approximately 30 mL of milk was collected from each sow at 2 h, 12 h, and 1 d after parturition and stored at −20 °C for further analysis. Venous blood was retrieved from the ear vein of sows at 1 d after delivery and on the day of weaning before feeding. Blood was then centrifuged (15 min at 3000 r/min at 4 °C), and serum samples were collected and stored at −20 °C for further analysis. The number of stillbirths, birth weight and weaning weight of the litter, mortality, and on-site weight of piglets were recorded.

### 2.4. Samples Analyses

Milk fat, protein, and lactose were measured using an automatic analyzer (MILKYWAYCP2, Institute of Food Science and Fermentation Engineering, Zhejiang University). The level of milk immunoglobin G (IgG), serum glucose (GLU), urea nitrogen (UREA), calcium (Ca), and nicotinic acid were determined using colorimetry (IgG and nicotinic acid kits: Beijing Sino-UK Institute of Biological Technology; others kits: BioSino Bio—Technology & Science Inc., Beijing, China) using an automatic biochemical analyzer (Hitachi 7160. Tokyo, Japan). The level of serum sodium was determined using the atomic absorbency method and a sodium-potassium analyzer (GB-7, Shanghai Xunda Medical Instrument Co., Ltd. Shanghai, China). The level of serum interleukin-6 (IL-6), TNF-α, Cortisol (COR), Triiodothyronine (T3), and tetraiodothyronine (T4) were determined using radioimmunoassay (kits: Beijing Sino-UK Institute of Biological Technology. Beijing, China) and an automatic radioactivity counting instrument (r-911, University of Science and Technology of China Industrial Corporation, Beijing, China).

### 2.5. Statistical Analysis

Statistical analyses were performed using the General Linear Model (GLM) program of SPSS 25.0 software (SPSS Inc., Chicago, IL, USA), arranged as a 2 × 2 factorial design in a completely randomized manner, with the environment condition and dietary MOS level as the main factors, and their interactions. Post hoc testing was conducted using Duncan’s multiple comparison tests. *p* < 0.05 was considered to indicate significance, and 0.05 < *p* < 0.1 was considered to indicate a trend. All data are presented as mean ± SEM [24].

## 3. Results

### 3.1. Environmental Conditions

In this experiment, the temperature range of the HS condition was 29–35 °C, with a daily average of 31.56 ± 1.22 °C and a THI of 84.68 ± 1.35 (81.96–87.81). The temperature range of the AC condition was 22–26 °C, with a daily average of 23.49 ± 0.72 °C and THI of 70.97 ± 0.18 (69.03–72.84). Over the experimental period, the humidity in the HS and AC groups was 67.66 ± 1.12 and 61.74 ± 1.10, respectively. Relative humidity is expressed as a percentage (%) (Figure 1).

### 3.2. Effect of MOS Supplementation on Body Condition of Sows

To confirm the effect of MOS on sow body condition, the body condition and farrowing duration of pigs were determined (Table 2). There were no significant differences in sow ADFI (average daily feed intake), body weight, backfat thickness, and farrowing duration between the HS and AC groups (*p* > 0.05). However, the weight loss of sows in the AC group was significantly lower than that of sows in the HS group (*p* < 0.01). Further, the weight loss of sows supplemented with MOS displayed a decreasing trend. Compared with that in the HS group, the weight loss of sows in the HM group decreased by 40.74%. Under HS conditions, the loss of backfat in the HM group displayed a downward trend and decreased by 33.29%, compared with that in the HS group. Nonetheless, we did not observe an interactive effect between MOS and HS on the body condition of sows.

### 3.3. Effect of MOS Supplementation on Reproductive Performances of the Sows and Growth Performances of Their Piglets

We determined the effect of MOS on the reproductive performance of sows; the results are presented in Table 3. The number of live litter and piglets born alive, average birth weight, average weaning weight, and survival rate of piglets did not significantly differ between the HS and AC groups (*p* > 0.05). Compared to the HS group, the average birth weight of the litter in the HM group increased by 18.79%. The number of piglets born alive for sows supplemented with MOS displayed an upward trend, compared with that in the HS group (*n* = 7.4), the number of piglets born alive in the HM group increased by 22.97% (*n* = 9.1). However, MOS and HS did not have an interactive effect on sow reproductive performance and growth performances of their piglets.

### 3.4. Effect of MOS Supplementation on Milk Composition of Sows

We determined the effect of dietary supplementation with MOS on the milk composition of sows (Table 4). Milk fat, lactose, and milk IgG did not significantly differ between the groups at 2 h after delivery. However, compared with sows subjected to the AC condition (AC group and AM group), the milk protein of shows subjected to the HS condition (HS group and HM group) was significantly decreased (*p* < 0.05). At 12 h after sow delivery, the milk protein and lactose of sows subjected to the HS condition (HS group and HM group) were significantly decreased (*p* < 0.01) compared with that of sows subjected to the AC condition (AC group and AM group). Further, milk IgG increased by 11.15% in the HS group compared with that in the HM group (*p* < 0.05). There were no significant differences in milk protein, milk fat, and lactose between sows in the different groups at 24 h after delivery. However, milk IgG was significantly higher in the HS group than the HM, AC, and AM groups (*p* < 0.05). Further, milk fat displayed a downward trend in the HS group compared with that in the HM group, with a decrease of 27.87%. MOS and HS were not found to have an interactive effect on the milk composition of sows.

### 3.5. Effect of MOS Supplementation on Serum Biochemical Indexes of Sows

We determined the effect of dietary MOS supplementation on the serum biochemical indexes of sows. The serum index parameters are presented in Table 5. On the first day after delivery, there were no significant differences in the levels of UREA, Ca, Na, GLU, Nicotinic acid, TNF-α, COR, T3, and T4 among the groups. However, the level of IL-6 significantly increased to 21.12% in the HS group compared with that in the HM group (*p* < 0.05). On the day of weaning, there were no significant differences in the levels of Ca, Na, Nicotinic acid, IL-6, COR, T3, and T4 among the groups. However, levels of UREA and GLU significantly decreased under HS condition (HS group and HM group) compared with that under AC condition (AC group and AM group). Further, TNF-α level significantly decreased in the AM group compared with the that in the other groups (HS, HM, and HS group) (*p* < 0.05). Compared with that in the AM group, TNF-α level in the AC group increased by 11.18%. However, MOS and HS did not have an interactive effect on the serum biochemical indexes of sows.

## 4. Discussion

MOS is widely used as a raw material in animal feed [25,26] as the use of prophylactic antibiotics in animal feed is banned worldwide [27,28]. Supplementation of piglet diets with MOS improved their growth performance and immunity [29,30]. However, the effect of MOS on the regulation of sows in HS environment is largely unclear. The current study sought to explore the effect of MOS on the condition of the body and the reproductive and lactation performance of sows subjected to HS.

In this study, the temperature under HS condition ranged from 29–35 °C, with a daily average of 31.56 ± 1.22 °C and THI of 84.68 ± 1.35 (81.96–87.81). The temperature and THI in the HS condition has been reported to be 30–35 °C and 82–86 [22,31], which were used to decide that the sows were experiencing HS. HS reduces pig growth performance and carcass fat quality, causing significant economic losses in the pig industry [32,33]. In the present study, the weight loss of sows in the HS group was significantly increased (*p* < 0.01). However, MOS and HS had no interactive effect on the body condition of sows. The weight loss and backfat thickness loss of sows supplemented with MOS displayed a decreasing trend. Compared with that in the HS group, the weight loss and backfat thickness loss in the HM group decreased by 40.74% and 33.29%, respectively. Backfat thickness loss during lactation has been reported to be positively associated with weaning-to-estrous intervals (WOI) [34]. Kim revealed that weight loss, backfat thickness loss, and WOI significantly increased in sows at high temperatures (28 °C) [35]. Further, sows housed in a cold room (23.9 °C) were found to display a decreasing trend in weight loss (*p* = 0.07) and backfat thickness (*p* = 0.07) loss compared to sows housed in the control room (29.4 °C) [36]. Through oral administration of MOS (200 mg·kg^−1^·d^−1^) to diet-induced obese mice for 4 weeks, Yan found that MOS significantly reduced body weight gain, insulin resistance, fatty liver, and inflammatory responses in obese mice, stimulated lipolysis, and inhibited lipogenesis in adipose tissues [37]. These results are consistent with those of the present study, as MOS tended to alleviate the loss of body weight and backfat thickness in sows under the HS condition.

Seasonal infertility is a major problem in the pig industry as the animals are affected by HS [38,39]. HS during gestation causes long-term developmental damage to the offspring of sows [40,41]. In this study, the average birth weight of the litter was significantly lower in the HS group than that in the other groups (HM, AC, and AM group). Furthermore, sow diet supplementation with MOS significantly increased the average birth weight of the litter under HS condition. Compared with that in the HS group, the average birth weight of the litter in the HM group increased by 18.79%. The birth weight of the litter is an important factor in pig production. Birth weight has a positive effect on colostrum intake, and contributes to a reduction in piglet mortality [42]. According to prior studies, the birth weights of piglets from sows exposed to HS during gestation was significantly lower than those born to sows exposed to neutral heat (1180 ± 50 vs. 1409 ± 59 g, respectively) [40]. Johnson revealed that intrauterine HS in sows could reduce the birth weight, weight gain, and reproductive efficiency of piglets [43]. To our knowledge, the effect of dietary MOS supplement on the reproductive performance of sows under HS has not been reported yet. However, dietary supplementation with 0.1% MOS had beneficial effects on the growth performance and nutrient digestibility and reduced diarrhea scores in weaned pigs [44]. According to Zhou, administering a combination diet of 1:1 AM/AP starch, 3% NSP, and 400 mg/kg MOS improved the growth performance and nutrient digestibility of piglets, increased butyric acid producing bacteria, and enhanced lipid metabolism [18]. These findings suggest that MOS supplementation in sow diets can improve the litter weight and piglet growth performance of sows under HS condition.

Lactating sows are susceptible to HS, and sow lactation weight loss significantly increases under HS conditions [45,46]. In the present study, we found that milk protein significantly decreased under HS condition in sows at 2 h and 12 h after delivery. Compared with that in the HM group, milk IgG in the HS group increased by 11.15% and 21.76% in sows at 12 h and 24 h after delivery. Recent studies suggest that HS is involved in the induction of tissue oxidative stress (OS). HS-induced OS may contribute to a decrease in milk protein content as OS promotes insulin resistance and apoptosis, and is negatively correlated with the synthesis of milk protein [47,48]. Westland revealed that the IgG concentration displayed an upward trend in prepartum dairy cows supplemented with 1.33% MOS (Control vs. MOS, 53.7 ± 5.8 vs. 42.7 ± 4.9) [49]. Calves fed colostrum supplemented with MOS had less efficient IgG uptake and lower 24-h serum IgG concentrations [23.9% (1.0); IgG = 24.0 (1.1) g/L] than control calves [30.4% (1.0); IgG = 30.8 (1.0) g/L] [50]. Although the effect of increasing dietary MOS supplement on milk protein and IgG in sows under HS conditions has not been exclusively reported, these results collectively suggest that dietary supplementation with MOS can reduce IgG and increase protein content in milk.

HS significantly alters metabolism, endocrine and immune physiology [51], gut barrier integrity, and gut barrier characteristics, which are important factors for the development of systemic immunity in the body [52,53]. In this study, HS significantly decreased serum UREA and GLU, but significantly increased IL-6. Dietary supplementation with MOS significantly reduced TNF-α under AC conditions. In fact, TNF-α decreased by 11.18% in the AM group relative to that in the AC group. IL-6 is a major pro-inflammatory cytokine in the IL family [54]. IL-6 and TNF-α play key roles in the pathogenesis of inflammatory diseases [55]. HS significantly decreased UREA in growing pigs [56]. Mendoza revealed that under HS, lipopolysaccharide (LPS), which is considered harmless during healthy animal challenge, aggravated the body’s inflammatory response, including a significant increase in IL-6 and IL-1β levels [43,57]. Dietary MOS supplementation significantly reduced TNF-α in broiler serum under HS condition [11], reduced oxidative stress and inflammatory response due to *Escherichia coli* infection, and significantly reduced TNF-α and NF-ϰB mRNA expression in liver tissue [58]. These results align with those of the present study and suggest that MOS can alleviate the inflammatory reaction caused by HS.

## 5. Conclusions

Overall, our study revealed the significant effect of HS on the body condition and lactation performance and their offspring of sows. Diets supplemented with 1 g/kg MOS tended to alleviate HS in sows. However, no significant interactive effects were found. Herein, HS was found to significantly increase sow weight loss, and significantly decrease milk protein, milk fat, and lactose. Meanwhile, supplementation with MOS reduced sow weight loss and backfat thickness loss, significantly increased average birth weight of the litter and milk protein, and decreased milk IgG, serum IL-6, and TNF-α of sows subjected to HS. Overall, our study revealed that a diet supplemented with 1 g/kg MOS has limited regulatory effect on HS in sows. However, the dosage and effect of MOS on sows warrants further investigation.

## Figures and Tables

**Figure 1 animals-12-01397-f001:**
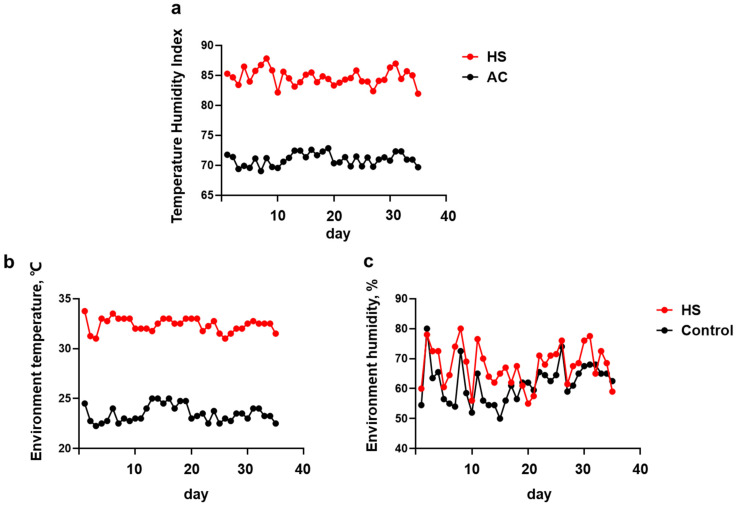
Temperature and humidity recordings obtained during the experiment. The environment THI (**a**), temperature (**b**), and humidity (**c**) were collected at the farm of Wuhan Jinying Animal Husbandry Co., Ltd. Data are presented as mean ± SEM.

**Table 1 animals-12-01397-t001:** Ingredients and formulated dietary nutrients (as-fed basis, %).

Item	Diet				
	Gestation	Lactation		Gestation	Lactation
Ingredients (%)			Nutrients		
Corn grain	63.00	63.00	DE (Mcal/kg)	3.14	3.26
Soybean meal	23.00	22.50	CP (%)	16.64	17.00
Wheat bran	10.00	7.00	Calcium (%)	0.78	0.84
Fish meal	-	1.50	Phos-available (%)	0.30	0.34
Soybean oil	-	2.00	CF (%)	3.18	2.96
Premix ^1^	4.00	4.00	DL-Lys (%)	0.98	1.02
QTotal	100	100	DL-Met (%)	0.25	0.27

^1^ Provided the following per kg of premix: Lys 40 g; Ca 170 g; Phos-available 40 g; NaCl 125 g; choline 4000 mg; organic chromium 5 mg, vitamin A 3.4 × 10^6^ IU; vitamin D3 6 × 10^5^ IU; vitamin E 2500 mg; vitamin K3 120 mg; vitamin B1 50 mg; vitamin B2 180 mg; vitamin B6 90 mg; vitamin B12 630 ug; vitamin H 12,000 ug; pantothenic acid 630 mg; folate 100 mg; niacin 1000 mg; Fe 3750 mg; Mn 1000 mg; Cu 562.5 mg; Zn 3250 mg; I 25 mg; and Se 12.5 mg.

**Table 2 animals-12-01397-t002:** Effects of dietary supplementation with MOS on the body condition of sows subjected to HS or AC.

Item	HS		AC		SEM	*p*-Values		
	HS	HM	AC	AM		MOS	HS	Interaction
ADFI (kg)	3.45	3.39	3.58	3.80	0.14	0.542	0.790	0.391
Body weight (kg)								
Entry	250.42	231.71	241.89	238.41	3.74	0.087	0.893	0.207
Wean	223.78	212.86	230.50	229.00	4.21	0.330	0.115	0.424
Change	−26.64	−18.85	−11.38	−9.41	1.79	0.158	<0.001	0.408
Backfat depth (mm)								
Entry	19.21	16.81	17.39	17.35	0.40	0.128	0.442	0.139
Wean	16.79	17.14	17.33	17.82	0.42	0.817	0.330	0.734
Change	−2.43	−0.07	−0.05	0.47	0.39	0.067	0.063	0.241
Duration of farrowing (h)	5.61	4.96	4.89	3.86	0.30	0.166	0.135	0.745

HS conditions: temperature: 31.56 ± 1.22 °C (29.25–35.75 °C), temperature-humidity index: 84.68 ± 1.35 (81.96–87.81); AC conditions: temperature: 23.49 ± 0.72 °C (26.90–30.20 °C), temperature-humidity index: 70.97 ± 0.18 (69.03–72.84). Data are expressed as mean ± SEM (*n* = 20) and were analyzed using the GLM program.

**Table 3 animals-12-01397-t003:** Effects of dietary supplementation with MOS on the reproductive performance of sows under the HS or AC conditions.

Item	HS		AC		SEM	*p*-Values		
	HS	HM	AC	AM		MOS	HS	Interaction
Number of litter size alive	7.8	9.1	9.2	9.0	0.21	0.189	0.129	0.314
Number of piglets born alive	7.4	9.1	9.0	8.9	0.22	0.061	0.088	0.572
Average birth weight of litter (kg)	12.1	14.9	14.7	15.0	0.41	0.049	0.081	0.621
Average birth weight of piglets (kg)	1.5	1.6	1.6	1.7	0.03	0.285	0.323	0.525
Average weaning weight of piglets (kg)	5.6	6.3	6.1	6.5	0.18	0.191	0.351	0.947
Survival rate of piglets (%)	0.97	0.96	0.98	0.98	0.01	0.566	0.288	0.563

HS conditions: temperature: 31.56 ± 1.22 °C (29.25–35.75 °C), temperature-humidity index: 84.68 ± 1.35 (81.96–87.81); AC conditions: temperature: 23.49 ± 0.72 °C (26.90–30.20 °C), temperature-humidity index: 70.97 ± 0.18 (69.03–72.84). Data are expressed as mean ± SEM (*n* = 20) and were analyzed using the GLM program. Different superscript letters in the table indicate significant difference (*p* < 0.05).

**Table 4 animals-12-01397-t004:** Effects of dietary supplementation with MOS on the milk composition of sows subjected to the HS and AC conditions.

Item		HS		AC		SEM	*p*-Values		
		HS	HM	AC	AM		MOS	HS	Interaction
2 h after delivery	Milk protein (%)	7.67	7.50	8.23	7.78	0.09	0.079	0.015	0.296
	Milk fat (%)	3.96	3.88	2.89	3.34	0.11	0.356	<0.001	0.710
	Lactose (%)	12.23	12.47	13.18	12.46	0.17	0.471	0.172	0.255
	Milk IgG (g/L)	57.26	55.59	56.67	59.04	5.20	0.481	0.095	0.653
12 h after delivery	Milk protein (%)	6.79	7.21	8.02	7.76	0.11	0.641	<0.001	0.053
	Milk fat (%)	4.08	4.13	3.84	3.28	0.11	0.224	0.011	0.142
	Lactose (%)	10.56	11.31	12.40	12.17	0.20	0.455	<0.001	0.164
	Milk IgG (g/L)	45.94	41.33	43.22	45.48	3.10	0.068	0.072	0.246
24 h after delivery	Milk protein (%)	6.36	6.77	6.35	6.04	0.14	0.874	0.235	0.246
	Milk fat (%)	5.02	6.96	4.71	4.66	0.28	0.087	0.020	0.071
	Lactose (%)	10.30	9.95	10.11	9.72	0.19	0.365	0.599	0.954
	Milk IgG (g/L)	40.74	33.46	31.06	34.86	3.50	0.078	0.048	0.231

HS conditions: temperature: 31.56 ± 1.22 °C (29.25–35.75 °C), temperature-humidity index: 84.68 ± 1.35 (81.96–87.81); AC conditions: temperature: 23.49 ± 0.72 °C (26.90–30.20 °C), temperature-humidity index: 70.97 ± 0.18 (69.03–72.84). Data are expressed as mean ± SEM (*n* = 20) and were analyzed using the GLM program. Different superscript letters in the table indicate significant difference (*p* < 0.05).

**Table 5 animals-12-01397-t005:** Effects of dietary supplementation with MOS on the serum index of sows subjected to HS or AC conditions.

Item		HS		AC		SEM	*p*-Values		
		HC	HM	AC	AM		MOS	HS	Interaction
1 day after delivery	UREA (mmol/L)	5.23	5.63	5.28	5.07	0.11	0.480	0.422	0.296
	Ca (mmol/L)	2.58	2.60	2.66	2.63	0.02	0.799	0.056	0.710
	Na (mmol/L)	147.77	149.54	147.05	153.11	1.28	0.074	0.385	0.255
	GLU (mmol/L)	4.35	4.61	4.75	4.73	0.07	0.605	0.163	0.262
	Nicotinic acid (ug/mL)	24.27	24.03	25.40	26.03	0.96	0.613	0.244	0.539
	IL-6 (pg/mL)	130.22	102.72	120.98	117.81	3.97	0.054	0.707	0.123
	TNF-α (pg/mL)	68.65	66.26	68.41	66.02	0.79	0.141	0.880	1.000
	COR (ng/mL)	151.72	164.50	164.46	166.32	2.82	0.197	0.199	0.334
	T3 (ng/mL)	0.61	0.57	0.59	0.59	0.01	0.234	0.893	0.373
	T4 (ng/mL)	41.29	41.31	41.75	41.60	0.91	0.975	0.844	0.963
Day of weaning	UREA (mmol/L)	5.07	5.47	6.23	6.30	0.86	0.219	<0.001	0.385
	Ca (mmol/L)	2.54	2.56	2.44	2.51	0.13	0.252	0.055	0.485
	Na (mmol/L)	146.57	147.48	148.51	148.34	11.30	0.908	0.660	0.865
	GLU (mmol/L)	4.03	3.76	4.54	4.12	0.81	0.024	0.005	0.642
	Nicotinic acid (ug/mL)	26.15	27.28	28.45	30.38	6.27	0.373	0.118	0.815
	IL-6 (pg/mL)	113.02	113.92	120.06	114.73	9.72	0.295	0.188	0.075
	TNF-α (pg/mL)	64.80	63.81	65.41	58.83	6.10	0.018	0.164	0.077
	COR (ng/mL)	132.90	125.98	135.44	123.94	17.93	0.063	0.959	0.639
	T3 (ng/mL)	0.59	0.61	0.61	0.60	0.62	0.998	0.763	0.362
	T4 (ng/mL)	40.73	41.93	43.43	42.83	5.46	0.845	0.237	0.552

HS conditions: temperature: 31.56 ± 1.22 °C (29.25–35.75 °C), temperature-humidity index: 84.68 ± 1.35 (81.96–87.81); AC conditions: temperature: 23.49 ± 0.72 °C (26.90–30.20 °C), temperature-humidity index: 70.97 ± 0.18 (69.03–72.84). Data are expressed as mean ± SEM (*n* = 20) and were analyzed using the GLM program. Different superscript letters in the table indicate significant difference (*p* < 0.05).

## Data Availability

The data used and analyzed in the current study are available from the corresponding author on reasonable request.

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
