# Peer review of "Effect of Dietary Supplementation with Mannose Oligosaccharides on the Body Condition, Lactation Performance and Their Offspring of Heat-Stressed Sows"

_animals, 2022, doi:10.3390/ani12111397_

Round 1

Reviewer 1 Report

The major concerns are with the hypothesis/rationale of the study and how the authors have interpreted the results of the main effect of MOS and heat stress as if they are interactive effects (that is, MOS x heat stress effects) throughout the manuscript. This has rendered some of the discussion points and the conlucsion invalid. In what follows, I have provided specific comments to support the major concerns. 

Abstract: The conclusion is false. There were mostly main effects of MOS and HS and a few tendencies (milk protein, milk fat, IL6, and TNF alpha). This suggests MOS was not able to elicit any beneficial effect on sows when exposed to heat stress. 

Introduction: Introduction was not informative enough as the mode of action of MOS to elicit the positive effects within the context of heat stress has been clearly articulated in this paper. The authors focused more on MOS effect as a prebiotic and on gut colonization but what has this got to do with managing summer heat stress in lactation sows? Are those immune bio-markers exclusively associated with heat stress or general immune system perturbation?

Statistical analysis: GLM is not appropriate for the statistical analysis of factorial designs as it is not able to handle within-subject effects. The authors should consider using LMM or GLMM for the statistical analysis. Also, the post hoc analysis should be for significant interactive effects, not main effects as done throughout the manuscript.  There were no significant interactive effects in this study, except for a few tendencies. Therefore, the use of superscripts is rather misleading. Especially, when you have two levels of each factor in a factorial design, a significant main effect is easy for the reader to see from the table without the need for mean separation.   

L162-163: This description is for interactive effect but this was not observed in this study. Also, it was sow BW at entry that tended to be improved by MOS. However, BW at entry includes wt of placenta and total litter weight (live born, stillbirth, and mummies). Unless, all this information is presented in the manuscript, MOS effect on the entry BW cannot be established.    

L179-179: Not interactive effect, only the main effects of MOS and HS

L234-236: This looks like a description of an interactive effect. However, there we no significant interactive effects between MOS and heat stress. This suggests MOS did not have any positive effects on sow BW and backfat loss that were under Heat stress.  

Conclusion: Not reflecting the data. There were mostly main effects of MOS or HS on the parameters with a few tendencies on milk fat, milk protein, and serum IL6 and TNF alpha.  Also, be specific with what was measured for the conclusion. Reproductive performance is a broad word and this paper only looked at litter characteristics at birth and piglet preweaning performance.  

Author Response

We highly appreciate detailed, constructive, and insightful comments from peer reviewer. We have carefully considered all comments, concur with most of them, and have incorporated reviewers’ suggestions in the revised manuscript. All changes are highlighted in yellow. We believe the quality of the revised manuscript is much strengthened as a result. The following is our point-by-point response to reviewers’ comments:

Reviewer #1

The major concerns are with the hypothesis/rationale of the study and how the authors have interpreted the results of the main effect of MOS and heat stress as if they are interactive effects (that is, MOS x heat stress effects) throughout the manuscript. This has rendered some of the discussion points and the conlucsion invalid. In what follows, I have provided specific comments to support the major concerns.

Response: Thank you for your suggestion and we apologize for the inadequate explanation of our experiment and data. We have corrected that “Based on our results, dietary supplementation with MOS tended to alleviate HS in sows compared with the control; however, no significant interactive effects were found.” in line 14-16 of the article

 Abstract: The conclusion is false. There were mostly main effects of MOS and HS and a few tendencies (milk protein, milk fat, IL6, and TNF alpha). This suggests MOS was not able to elicit any beneficial effect on sows when exposed to heat stress.

Response: Thank you for your suggestion. We have corrected that “In conclusion, HS significantly affected the body condition, lactation performances and their off-spring of sows. However, dietary supplementation with 1 g/kg MOS did not result in statistically significant changes.” In line 32-34 of the article

Introduction: Introduction was not informative enough as the mode of action of MOS to elicit the positive effects within the context of heat stress has been clearly articulated in this paper. The authors focused more on MOS effect as a prebiotic and on gut colonization but what has this got to do with managing summer heat stress in lactation sows? Are those immune bio-markers exclusively associated with heat stress or general immune system perturbation?

Response: Thank you for your comment. I apologize for not expressing the correct logic. The essence of MOS is a prebiotic, the main mechanism of MOS is to improve the intestinal environment. So the mechanism of MOS to relieve heat stress in sows may be through improving the intestinal environment to improve nutrient absorption and immune stress under heat stress of the organism. According to your suggestion, we have added relevant literature and content “Supplementation of MOS at 250 mg/kg in the diet was effective in improving broiler growth performance (ADG, ADFI and feed conversion ratio) [11]. Supplementing MOS in the diet improved the quality of colostrum, and promoted piglet growth performance and innate immunity in lactating and nursery piglets [12]. The increase in ADG was due to the promotion of MOS on nutrients digestibility, gut microbiology and barrier integrity, Oxidative state and animal immunity. He et al. showed that gut micro-biota composition responding to HS could be considered as biomarkers in monogastric animals [13].” “Diet supplemented with MOS may contribute to the maintenance of intestinal integrity and nutrient digestion and absorption in the intestine after weaning” In line 56-63 and 70-71 of the article

Statistical analysis: GLM is not appropriate for the statistical analysis of factorial designs as it is not able to handle within-subject effects. The authors should consider using LMM or GLMM for the statistical analysis. Also, the post hoc analysis should be for significant interactive effects, not main effects as done throughout the manuscript.  There were no significant interactive effects in this study, except for a few tendencies. Therefore, the use of superscripts is rather misleading. Especially, when you have two levels of each factor in a factorial design, a significant main effect is easy for the reader to see from the table without the need for mean separation.

Response: Thank you for your suggestion. This study is a 2×2 factorial experiment, so we used the GLM. I don’t know the use of superscripts before. According to your suggestion, I have corrected the superscripts in tables, and deleted the superscripts when there were no significant interactive effects.

L179-179: Not interactive effect, only the main effects of MOS and HS

Response: Thank you for your suggestion and we apologize for the inadequate explanation of our experiment and data. We have corrected that “we did not observe an interactive effect between MOS and HS on the body condition of sows.” “however, MOS and HS did not have an interactive effect on sow reproductive performance and growth performances of their piglets.” “MOS and HS were not found to have an interactive effect on the milk composition of sows” “however, MOS and HS did not have an interactive effect on the serum biochemical indexes of sows.” in line 182-183, line 199-200, line 219-220 and line 239-240 of the article

L234-236: This looks like a description of an interactive effect. However, there we no significant interactive effects between MOS and heat stress. This suggests MOS did not have any positive effects on sow BW and backfat loss that were under Heat stress. 

Response: Thank you for your suggestion and we apologize for the inadequate explanation of our experiment and data. According to your suggestion, we added the content that “however, MOS and HS had no interactive effect on the body condition of sows.” in line 259 of the article

Conclusion: Not reflecting the data. There were mostly main effects of MOS or HS on the parameters with a few tendencies on milk fat, milk protein, and serum IL6 and TNF alpha.  Also, be specific with what was measured for the conclusion. Reproductive performance is a broad word and this paper only looked at litter characteristics at birth and piglet preweaning performance. 

Response: Thank you for your suggestion, we apologize for the inadequate explanation of our experiment and data. According to your suggestion, I corrected that “our study revealed the significant effect of HS on the body condition and lactation performance and their offspring of sows. Diets supplemented with 1 g/kg MOS tended to alleviate HS in sows; however, no significant interactive effects were found. Herein, HS was found to significantly increase sow weight loss, and significantly decrease milk protein, milk fat, and lactose” “our study revealed that a diet supplemented with 1 g/kg MOS has limited regulatory effect on HS in sows. However, the dosage and effect of MOS on sows warrant further investigation”. in line 328-332 and 335-337 of the article

Reviewer 2 Report

There is not a clear hypothesis

Introduction don´t justify the interest of the study. Heat stress related results in broilers and the likely effects on microbiota or immunity don´t justify the basis of this study on performance, body condition and reproductive and lactation performance

Number of sows appears to be very low observe differences, and the distribution of primiparous and multiparous among treatments is not well described.

Nutrient composition between gestation and lactation don´t appears to follow NRC requirements

Air conditioning may assure a constant temperature, but environmental temperature may change in uncontrolled rooms. Are those Maximum Temperatures? (what is it the range, number of hour above 30ºC?)

Moreover, there are a number of results which appears not to be credible

Average voluntary intake during a 21d lactation < 4kg/d is not credible for sows of 240-250kg and a litter weight gain of 40-45 kg in 21 days

BW entry for HS_HM  were lower than HS_HS which may indicate a biased distribution of animals.

Backfat depth did not change for HS_HM during lactation which poses doubts about the experimental data in general

Author Response

We highly appreciate detailed, constructive, and insightful comments from peer reviewer. We have carefully considered all comments, concur with most of them, and have incorporated reviewers’ suggestions in the revised manuscript. All changes are highlighted in yellow. We believe the quality of the revised manuscript is much strengthened as a result. The following is our point-by-point response to reviewers’ comments:

Reviewer #2

Introduction don´t justify the interest of the study. Heat stress related results in broilers and the likely effects on microbiota or immunity don´t justify the basis of this study on performance, body condition and reproductive and lactation performance

Response: Thank you for your suggestion, but I have justified the basis of this study on performance, body condition and reproductive and lactation performance in line 39-48 of the article “HS leads to low farrowing rates of breeding sows in summer, researches have con-firmed that hot season reduce sperm motility and concentration in boars, resulting in lower farrowing rates in summer mating sows [1, 2]. In addition, sows mated in summer are more prone to early pregnancy abortion than in other seasons [3]. HS increases carcass fat in offspring of sows bred in summer, as it reduces fetal muscle fiber development during pregnancy [4]. HS reduces feed intake and growth rates in summer pigs [5], and leads to the reduction of milk production in lactating sows [6]. These three adverse results caused by HS were identified important factors affecting the pig industry [7].”

Number of sows appears to be very low observe differences, and the distribution of primiparous and multiparous among treatments is not well described.

Response: Thank you for your comment, I am sorry that the experiment of 80 sows is already the limit in Wuhan city.

Nutrient composition between gestation and lactation don´t appears to follow NRC requirements

Response: Thank you for your suggestion and we apologize for the inadequate explanation of our experiment. The diets we use have been confirmed to be commercial standards, we have corrected it in line 118 of the article

Air conditioning may assure a constant temperature, but environmental temperature may change in uncontrolled rooms. Are those Maximum Temperatures? (what is it the range, number of hour above 30ºC?)

Response: Thank you for your comment, the experiment was conducted in the hot July. I am sorry that the temperature is recorded using an average of three time periods per day, we don’t record every hour temperature.

Moreover, there are a number of results which appears not to be credible

Response: Thank you for your comment, I am sorry that the data of the sow experiment is difficult to collect, but the data in the article is credible and real.

Average voluntary intake during a 21d lactation < 4kg/d is not credible for sows of 240-250kg and a litter weight gain of 40-45 kg in 21 days

Response: Thank you for your comment, but in this study, sows were limited to 2.5–4 kg/d of feeding from entry until day 3 after farrowing and then fed ad libitum as previously described, the content in line 115-116 of the article.

BW entry for HS_HM were lower than HS_HS which may indicate a biased distribution of animals.

Response: Thank you for your suggestion and we apologize for the inadequate explanation of our experiment. The distribution of animals is that “Landrace ×Large white, primiparous (n=28)/ multiparous (n=52)” in line 88-89 of the article

Backfat depth did not change for HS_HM during lactation which poses doubts about the experimental data in general

Response: Thank you for your comment. I am sorry the data of the sow experiment is difficult to collect, but the data in the article is credible and real.

Reviewer 3 Report

1- Introduction: development of action mechanism of MOS

2- line 76 : primiparity and multiparity are terms to woman - use primiparous and multiparous from sows

3- Table 110: Lys (%) Met (%) - concrete Lys total and Met total or digestibility?

4- Line 178 y 180 - write Number vs number

5- Line 207 - 211 - 287 and Table 5 Line 217 - UREA vs urea or Urea

Author Response

We highly appreciate detailed, constructive, and insightful comments from peer reviewer. We have carefully considered all comments, concur with most of them, and have incorporated reviewers’ suggestions in the revised manuscript. All changes are highlighted in yellow. We believe the quality of the revised manuscript is much strengthened as a result. The following is our point-by-point response to reviewers’ comments:

Reviewer #3

1- Introduction: development of action mechanism of MOS

Response: Thank you for your suggestion, I have added the content that “Supplementation of MOS at 250 mg/kg in the diet was effective in improving broiler growth performance (ADG, ADFI and feed conversion ratio) [11]. Supplementing MOS in the diet improved the quality of colostrum, and promoted piglet growth performance and innate immunity in lactating and nursery piglets [12]. The increase in ADG was due to the promotion of MOS on nutrients digestibility, gut microbiology and barrier integrity, Oxidative state and animal immunity. He et al. showed that gut micro-biota composition responding to HS could be considered as biomarkers in monogastric animals [13].” “Diet supplemented with MOS may contribute to the maintenance of intestinal integrity and nutrient digestion and absorption in the intestine after weaning” In line 56-63 and 70-71 of the article.

2- line 76 : primiparity and multiparity are terms to woman - use primiparous and multiparous from sows

Response: Thank you for your suggestion, we have corrected that “(Landrace ×Large white, primiparous (n=28)/ multiparous (n=52))” in line 86 of the article

3- Table 110: Lys (%) Met (%) - concrete Lys total and Met total or digestibility?

Response: Thank you for your suggestion, we have confirmed and corrected that “DL-Lys” and “DL-Met” in table 1.

4- Line 178 y 180 - write Number vs number

Response: Thank you for your comment, we have corrected that “The Number of piglets born alive of sows added to MOS showed an upward trend, compared with HS group (n=7.4), the Number of piglets born alive in HM group in-creased by 22.97% (n=9.1)” in line 196-199 of the article

5- Line 207 - 211 - 287 and Table 5 Line 217 - UREA vs urea or Urea

Response: Thank you for your suggestion, we have unified the correction to UREA in line 318 of the article
